# Prognostic Implications and Predictors of Mitral Regurgitancy Reduction After Transcatheter Aortic Valve Implantation

**DOI:** 10.3390/medicina60122077

**Published:** 2024-12-18

**Authors:** Murat Can Güney, Hakan Süygün, Melike Polat, Hüseyin Ayhan, Telat Keleş, Zeynep Şeyma Turinay Ertop, Engin Bozkurt

**Affiliations:** 1Medicana International Ankara Hospital, Department of Cardiology, Faculty of Medicine, Atılım University, Söğütözü, 2176. Sk. No: 3, Çankaya 06510, Turkey; melike.polat@medicana.com.tr; 2Department of Cardiology, Faculty of Medicine, Karamanoglu Mehmetbey University, Karaman Training and Research Hospital, Karaman 70110, Turkey; hakansuygun@kmu.edu.tr; 3Department of Cardiology, Faculty of Medicine, University of Health Sciences Gulhane, Ankara Bilkent City Hospital, Ankara 06800, Turkey; huseyin.ayhan@sbu.edu.tr; 4Department of Cardiology, Faculty of Medicine, Ankara Yıldırım Beyazıt University, Ankara Bilkent City Hospital, Ankara 06800, Turkey; telat.keles@ybu.edu.tr; 5Department of Cardiology, Medicana International Ankara Hospital, Ankara 06530, Turkey; zeynep.ertop@medicana.com.tr (Z.Ş.T.E.); engin.bozkurt@medicana.com.tr (E.B.)

**Keywords:** TAVI, mitral regurgitancy reduction, mortality

## Abstract

*Background*: Mitral regurgitation (MR) is a common condition observed in patients undergoing transcatheter aortic valve implantation (TAVI) for the treatment of aortic stenosis (AS). However, the impact of TAVI on MR outcomes and the factors predicting MR improvement remains uncertain. Understanding these predictors can enhance patient management and guide clinical decisions. *Methods*: This retrospective cohort study included 156 patients with moderate to severe MR undergoing TAVI. MR severity was assessed via echocardiography at baseline, as well as 6 months and 1 year after TAVI. Patients were divided into groups based on MR reduction: no improvement or worsening, one-degree improvement, and at least two-degree improvement. Clinical, echocardiographic, and procedural characteristics were evaluated as predictive factors for MR improvement after TAVI. *Results*: MR reduction occurred in 68% of patients at 6 months and 81% at 1 year. Factors predicting a reduction of two grades or more in MR severity included lower baseline LVEDD (OR = 1.345, 95% CI: 1.112–1.628, *p* = 0.002) lower baseline LA (OR = 1.121, 95% CI: 1.015–1.237, *p* = 0.024), lower baseline LVMI (OR = 1.109, 95% CI: 1.020–1.207, *p* = 0.024), and higher baseline EF levels (OR = 1.701, 95% CI: 1.007–2.871, *p* = 0.047). No significant association was found between MR reduction at 6 months and one-year mortality. (*p* = 0.65). *Conclusions*: Baseline echocardiographic parameters are valuable in predicting MR improvement post-TAVI, with LVMI emerging as a novel predictor. However, MR reduction did not independently predict survival, underscoring the need for further research to optimize patient selection and management strategies in TAVI candidates.

## 1. Introduction

Mitral regurgitation (MR) is a common concomitant valvular pathology in patients with aortic stenosis (AS) [1]. In patients undergoing transcatheter aortic valve implantation (TAVI), the prevalence of moderate to severe MR has been reported to range from 22% to 48% [2]. However, most research and clinical trials primarily focus on single-valve diseases, often excluding patients with multivalvular conditions. This limitation complicates the decision-making process for the treatment of AS and MR, as these valvular pathologies closely interact hemodynamically. MR can reduce anterograde flow, thereby lowering the aortic valve gradient. Conversely, AS can worsen the severity of MR by increasing left ventricular (LV) afterload. Consequently, improvement in MR is often expected due to the hemodynamic changes and structural remodeling that follow aortic valve intervention [3,4].

TAVI in multivalvular disease also carries high procedural complication risks, highlighting the need for meticulous planning and patient selection in complex cases [5]. The determination of whether to treat a mitral valve concurrently during aortic valve procedures remains unresolved. As unnecessary mitral interventions during surgical or interventional aortic valve replacement can elevate the risks of mortality and morbidity, identifying clinical predictors of persistent MR is highly beneficial [6]. The aim of this study was to assess the prognostic significance of baseline MR and its reduction following TAVI while identifying potential demographic, laboratory, echocardiographic, and procedural factors that may predict MR severity improvement post-TAVI.

## 2. Methods

### 2.1. Patient Population

This study is a retrospective analysis of data collected from a dedicated database of patients with symptomatic severe AS (aortic valve area < 1 cm^2^ or aortic valve area index < 0.6 cm^2^/m^2^) who underwent TAVI between October 2011 and May 2023 at two centers. Among 615 consecutive patients treated with TAVI at the participating centers, 206 patients were detected to have echocardiographic criteria for moderate to severe MR, and 156 patients were included in the outcome analysis after excluding those with missing clinical or echocardiographic data at baseline or follow-up visits. Both participating centers in this study are tertiary reference centers for valvular heart diseases, and a significant proportion of our patients are referred from other cities. As a result, many of these patients continue their follow-up care at their local hospitals, limiting our access to comprehensive follow-up data. This limitation is the primary reason for the large number of patients excluded from the study due to missing data. Patients with previous mitral valve surgery were excluded. The study has received approval from the local ethics committee and adheres to the Declaration of Helsinki.

### 2.2. Data Collection

Patient data were collected using the institutional electronic database and individual patient charts. Mortality information was sourced from the National Health Service database and verified through hospital or telephone follow-ups. Follow-up data were collected during hospital visits at approximately 6 and 12 months post-procedure. Patients with missing substantial data from any of the visits were excluded from the analysis.

### 2.3. Echocardiography

Transthoracic echocardiography was performed with a Vivid 7 instrument (General Electric, Horten, Norway) and evaluated by local experienced echocardiographers according to American Society of Echocardiography (ASE) guidelines. LV diameters were obtained using M-mode imaging. Left ventricular ejection fraction (LVEF) was evaluated using the biplane Simpson method. Systolic pulmonary artery pressure was estimated through the peak velocity of tricuspid regurgitation. The severity of MR was determined by integrating measures from the color Doppler jet area, vena contracta, effective regurgitant orifice area, and proximal isovelocity surface area and classified on a scale from 0 to 4 (0 = non or trace MR, 1 = mild MR, 2 = moderate MR, 3 = moderate to severe MR, 4 = severe MR) according to ASE guidelines [7]. Patients with basal MR severity ≥ 2 are included in the study. The etiology of MR was classified as organic when MR resulted from prolapse, calcification, rheumatic, or endocarditic lesions and as functional when there were no mitral valve abnormalities, with the regurgitation being due to valve tethering and incomplete closure. Comprehensive transthoracic echocardiography was performed at 6-month and 1-year visits at the implanting center. Changes in MR grade at follow-up are grouped as no improvement or worsening, one-degree improvement in severity, or at least two-degree improvement in severity. Changes in LVEF left ventricular end-diastolic dimension (LVEDD) and left ventricular end-systolic dimension (LVESD) were calculated according to the baseline and follow-up echocardiographic data. Changes in echocardiographic parameters at each visit are calculated as delta left ventricular end-diastolic dimension (d-LVEDD), delta left ventricular end-systolic dimension (d-LVESD) and d-LVEF separately according to the following equations: d-LVDD (%) = [((LVDD at each visit after TAVI) − (Baseline LVDD))/(Baseline LVDD)] × 100 d-LVSD (%)= [((LVSD at each visit after TAVI) − (Baseline LVSD))/(Baseline LVSD)] × 100 d-LVEF (%) = [((LVEF at each visit after TAVI) − (Baseline LVEF))/(Baseline LVEF)] × 100. Relative wall thickness (RWT) was calculated as RWT = (2 × Posterior wall thickness (PWT)) ÷ LVEDD and left ventricular mass index (LVMI) as LVMI = 0.8 × (1.04 × ((LVEDD + PWT + IVS)3 − LVEDD3)) + 0.6 and indexed to body surface area (BSA).

### 2.4. TAVI Procedure

TAVI indication and procedural details were determined according to the assessment of the heart team, which consisted of a cardiologist, cardiac surgeon, and anesthesiologist. Contrast-enhanced multidetector computed tomography was used in all patients for procedure planning. The transfemoral approach was preferred in all cases. The procedure was carried out in a catheterization laboratory under general anesthesia or deep sedation under fluoroscopy guidance, as previously described in the literature [8]. Edwards SAPIEN XT and SAPIEN 3 valves were used in the majority of the cases (Edwards Lifesciences, Irvine, CA, USA).

## 3. Statistical Analysis

Data analysis was performed using IBM SPSS Statistics ver. 25 (IBM Corporation, Armonk, NY, USA) package program. Kolmogorov–Smirnov and Levene’s tests were used to investigate whether the assumptions of normal distribution and homogeneity of variances were met. Categorical data were expressed as numbers (n) and percentages (%), while quantitative data were given as mean ± SD or median (25th–75th) percentiles, where—appropriate. While the mean differences between groups (i.e., baseline MR Grade 2 vs. Grade 3–4) were compared by Student’s *t*-test, otherwise the Mann–Whitney U test was applied for comparisons of continuous variables for which parametric test assumptions were not met. Whether the differences between baseline and post-op 6th month regarding EF, LVEDD, and LVESD levels were statistically significant or not was evaluated by the Wilcoxon Sign Rank test. While the mean differences among groups (i.e., no change in MR grade, improved by no more than one degree and improved by at least two degrees at the end of the 6th month) were compared by one-way ANOVA, otherwise the Kruskal–Wallis test was applied for comparisons of continuous variables for which parametric test assumptions were not met. When the *p*-values from the Kruskal–Wallis test were statistically significant, the Dunn–Bonferroni multiple comparison test was used to know which group differs from which others. Unless otherwise stated, Pearson’s χ^2^ test was used in the analysis of categorical data. In all 2 × 2 contingency tables to compare categorical variables, the Continuity corrected χ^2^ test was used when one or more of the cells had an expected frequency of 5–25; otherwise, the Fisher’s exact test was used when one or more of the cells had an expected frequency of 5 or less. In the RxC (if at least one of the categorical variables in the row or column has more than two results) cross-tabulations, if the expected frequency was below 5 in at least a quarter of the cells, the Fisher–Freeman Halton test was used. Whether the associations between baseline MR grade and improvement in MR grade at the end of the 6th month with cumulative mortality were statistically significant or not were performed by univariate logistic regression analysis. In order to determine the best predictors of the improvement in mitral regurgitation grade at the end of the 6th month, it was investigated by Multinominal logistic regression analysis via a backward elimination procedure. Any variable whose univariable test had a *p*-value less than 0.25 was accepted as a candidate for the multivariable model. Odds ratios (*OR*), 95% confidence intervals (CI), and Wald statistics for each independent variable were also calculated. A *p*-value of less than 0.05 was considered statistically significant.

## 4. Results

In the current study, data from 155 patients were analyzed, with a median age of 79 years. Of these, 63 (40.6%) were male and 92 (59.4%) were female. Regarding the distribution of baseline MR severity, 87 patients (56.1%) had grade 2, 61 patients (39.4%) had grade 3, and 7 patients (4.5%) had grade 4 MR. In-hospital mortality occurred in 2 cases (1.3%), and the mortality rate at 6 months was 6.5% (10 patients). At the one-year follow-up, the cumulative mortality rate was 14.2% (22 patients).

From baseline to the 6-month follow-up (excluding deceased patients), MR severity decreased in 99 out of 145 patients (68%). Among the 133 patients who completed the one-year follow-up, a reduction in MR severity was observed in 109 patients (81%). Figure 1 presents a heat map of the frequency distributions of MR severity grades at baseline and at the 6-month follow-up.

For patients with baseline MR severity of grade 3–4, age, baseline LVEDD, and baseline left atrial (LA) dimensions were significantly higher compared to those with baseline grade 2 MR (*p* = 0.018, *p* = 0.037, and *p* = 0.042 respectively), while baseline EF levels were significantly lower (*p* = 0.026). No statistically significant differences were found in other examined characteristics between the groups (*p* > 0.05). Table 1 and Table 2 feature comparisons between groups with baseline MR severity grades 2 and grades 3–4. Descriptive statistics related to all the examined characteristics within the entire cohort are also provided.

According to the 6-month echocardiographic follow-up data, patients were divided into three groups: no improvement or worsening, one-degree improvement in severity, or at least two-degree improvement in severity. Regarding the data on potential factors influencing changes in MR severity at the 6-month follow-up, a statistically significant difference in baseline EF was observed between the groups (*p* = 0.012). This difference was due to higher baseline EF levels in the groups with MR severity improvement (either a one-grade improvement or at least a two-grade improvement) compared to the group with unchanged MR grade (*p* = 0.036 and *p* = 0.025, respectively). There was also a significant difference in baseline LVEDD levels between the groups, with lower baseline LVEDD levels observed in the groups with MR severity improvement (either a one-grade improvement or at least a two-grade improvement) compared to those with unchanged MR severity (*p* < 0.001 and *p* = 0.033, respectively). A statistically significant difference was observed in baseline LVESD levels (*p* < 0.001), with lower baseline LVESD levels found in the group with a one-grade improvement in MR severity compared to those with unchanged MR severity (*p* < 0.001). A significant difference was found in RWT levels; those with MR severity improvement at follow-up (either a one-grade improvement or at least two-grade improvement in MR severity) had higher RWT levels compared to those with unchanged MR (*p* = 0.026 and *p* = 0.022). There was a significant difference in LVMI levels (*p* < 0.001); groups with MR severity improvement at follow-up (either a one-grade improvement or at least two-grade improvement in MR severity) had lower LVMI levels compared to those with unchanged MR severity (*p* < 0.001 and *p* = 0.030). There was also a significant difference in baseline LA levels; those with MR severity improvement at follow-up (either a one-grade improvement or at least two-grade improvement in MR severity) had lower baseline LA levels compared to those with unchanged MR severity (*p* = 0.018 and *p* = 0.020). Table 3 and Table 4 present the results of univariate statistical analyses examining potential factors influencing or associated with changes in MR severity at the 6-month follow-up.

The next phase employed multinomial logistic regression analysis to identify the most determinative factor(s) predicting a decrease in MR severity from baseline to the 6-month follow-up. According to the stepwise backward elimination method (see Table 5), factors predicting a one-grade improvement in MR severity compared to the group with no improvement or worsening MR severity were lower baseline LVEDD (OR = 1.291, 95% CI: 1.097–1.520, *p* = 0.002) and baseline LA (OR = 1.124, 95% CI: 1.030–1.227, *p* = 0.009). Factors predicting a reduction of two grades or more in MR severity included lower baseline LVEDD (OR = 1.345, 95% CI: 1.112–1.628, *p* = 0.002) lower baseline LA (OR = 1.121, 95% CI: 1.015–1.237, *p* = 0.024), lower baseline LVMI (OR = 1.109, 95% CI: 1.020–1.207, *p* = 0.024), and higher baseline EF levels (OR = 1.701, 95% CI: 1.007–2.871, *p* = 0.047). In this model, baseline MR severity was included as a correction factor, revealing that patients with an MR grade of 3 or 4 demonstrated greater improvement in MR severity.

During the one-year follow-up, baseline MR severity was a statistically significant determinant of one-year mortality, with baseline MR grade 3 or 4 associated with a 3.235-fold increase in mortality risk (95% CI: 1.236–8.464, *p* = 0.017) compared to baseline MR grade 2.

When assessing the association between changes in MR severity at 6 months and one-year mortality, no statistically significant differences in mortality were identified between the groups with no improvement, worsening MR severity (OR = 0.558, 95% CI: 0.117–2.670, *p* = 0.465), or in one-grade improvement group (OR = 0.690, 95% CI: 0.173–2.752, *p* = 0.599) compared to at least two-grade improvement group. Moreover, regardless of whether baseline MR was grade 2 or 3–4, no cases that regressed to trace insufficiency by the 6-month follow-up were observed to have died. Table 6 represents the effects of baseline MR severity and changes in MR severity over a 6-month period on one-year cumulative survival.

## 5. Discussion

In this study, the baseline degree of mitral regurgitation (MR) was significantly associated with one-year mortality. However, while a statistically significant relationship between MR reduction following TAVI and one-year mortality could not be established, a non-significant trend toward lower mortality was observed in patients who experienced MR reduction. Additionally, a higher baseline MR grade (grade 3–4), along with higher baseline EF, lower baseline LVEDD, lower baseline LVMI, and lower baseline LA, were identified as significant predictors of MR reduction following TAVI. These findings suggest that specific baseline echocardiographic parameters may help identify patients more likely to benefit from MR improvement post-TAVI.

The observed one-year mortality rate of 14.2% aligns with the reported range in similar cohorts of patients undergoing TAVI with concomitant MR [9]. In our study, 68% of patients experienced a reduction in MR severity at six months, increasing to 81% at one year, which is comparable to the improvement rates reported in patients with MR in previous research [10].

Our study demonstrated a significant association between baseline MR grade and one-year mortality, aligning with existing literature that highlights the prognostic importance of baseline moderate to severe MR on post-TAVI outcomes [3,11,12]. However, unlike many studies that report a significant relationship between MR reduction after TAVI and improved postoperative survival, we did not find a statistically significant association between MR improvement at six months and one-year mortality [13,14]. This discrepancy may be due to relatively low levels of mortality in our cohort. Notably, none of the patients with trivial MR at six months experienced mortality at one year. This finding aligns with prior studies that report a connection between MR improvement and improved survival outcomes, suggesting that our results may reflect limitations in sample size or population-specific factors rather than an absence of association. Despite the lack of a significant relationship with mortality in our cohort, identifying predictors of MR reduction remains clinically important, as MR reduction has been shown to influence functional outcomes, quality of life, and hemodynamic recovery, which are critical metrics of success in TAVI procedures [15]. Additionally, these findings can support clinical decision-making by enabling effective risk stratification, which could enhance patient selection and potentially improve outcomes in this high-risk population.

Our study included a high prevalence of functional MR patients, accounting for 72% of the cohort. Research indicates that patients with functional MR are more likely to experience a reduction in MR severity after TAVI compared to those with organic MR, primarily due to improvements in left ventricular remodeling and a decrease in mitral annular dilatation [2,16]. However, our study did not demonstrate a significant difference in outcomes between MR etiology groups, which could be attributed to the relatively smaller proportion of patients with organic MR in our cohort. One possible explanation is that patients with organic MR and aortic stenosis are more frequently referred for double-valve surgery, potentially introducing a selection bias in our study.

In our study, we categorized patients with a reduction in MR after TAVI into two groups: those who experienced a one-degree improvement and those with at least a two-degree improvement in MR severity. Our analysis identified that higher LVEF, lower baseline LVEDD, lower baseline LVMI, and lower baseline LA size were significant predictors of greater MR reduction following TAVI. Our findings regarding the relationship between LA diameter and MR reduction are consistent with existing literature, which identifies LA size as a reliable predictor of outcomes in MR patients [17,18,19].

However, evidence regarding the predictive role of LVEDD and EF remains conflicting. Hekimian et al. found that MR decrease was more profound in patients with LV dysfunction and dilatation. But, they performed MR control at 7 days post-TAVI, and the majority of the patients had organic MR. Notably, to the best of our knowledge, our study is the first to demonstrate a significant association between higher baseline LVEF, lower baseline LVEDD, lower baseline LVMI, and MR reduction following TAVI. We evaluated data on MR improvement at 6 months in a cohort predominantly composed of patients with functional MR, providing a longer timeframe compared to some studies in the literature, which allowed us to eliminate the immediate unloading response observed right after TAVI and focus on the long-term left ventricular remodeling process [20]. Lower baseline LVMI likely reflects a less remodeled left ventricle with preserved myocardial integrity and function, which may contribute to a more effective left ventricular unloading and mitral valve coaptation post-TAVI. By emphasizing LVMI, our findings suggest that the degree of myocardial hypertrophy and structural remodeling before TAVI plays a pivotal role in determining the likelihood of MR reduction. This novel insight aligns with the concept that myocardium with less hypertrophic remodeling prior to the intervention is better equipped to adapt to hemodynamic changes, facilitating improved mitral valve function. Thus, LVMI serves as a promising addition to the preoperative assessment parameters, offering a more nuanced understanding of patient selection and outcomes following TAVI [21,22].

## 6. Study Limitations

This study has several limitations that should be acknowledged. First, as a retrospective cohort study, there is a potential for patient selection bias, which may have influenced the results. Selection bias can arise if the inclusion of patients is related to both the exposure and the outcome of interest, potentially affecting the reliability of the findings. Second, the assessment of MR severity was performed using both semiquantitative methods and quantitative evaluations, but we lack data on the latter. The absence of a core laboratory for echocardiographic evaluation further introduces variability, as measurements were not standardized across a central expert review but rather subject to inter-observer variation. This variability could affect the consistency and accuracy of MR severity grading across the study cohort. Additionally, there is a significant imbalance in patient numbers between the two groups of MR etiology, as patients with organic MR were more frequently referred for surgery. Finally, our analysis predominantly included patients with moderate MR and fewer cases of severe MR, which may limit the generalizability of our findings to patients with more advanced MR severity.

## 7. Conclusions

In conclusion, this study reveals that baseline MR severity was associated with long-term mortality. Our findings emphasize that the decrease in MR following TAVI did not independently predict one-year survival outcomes, which is in contrast with some reports in the literature. Notably, our study identified higher EF, lower LVEDD, lower LVMI, and lower LA size as significant predictors of MR improvement, contributing new insights, especially regarding the role of LVMI. These findings suggest that targeted patient selection and careful pre-procedural evaluation can enhance clinical outcomes and guide management strategies in TAVI patients with concomitant MR.

## Figures and Tables

**Figure 1 medicina-60-02077-f001:**
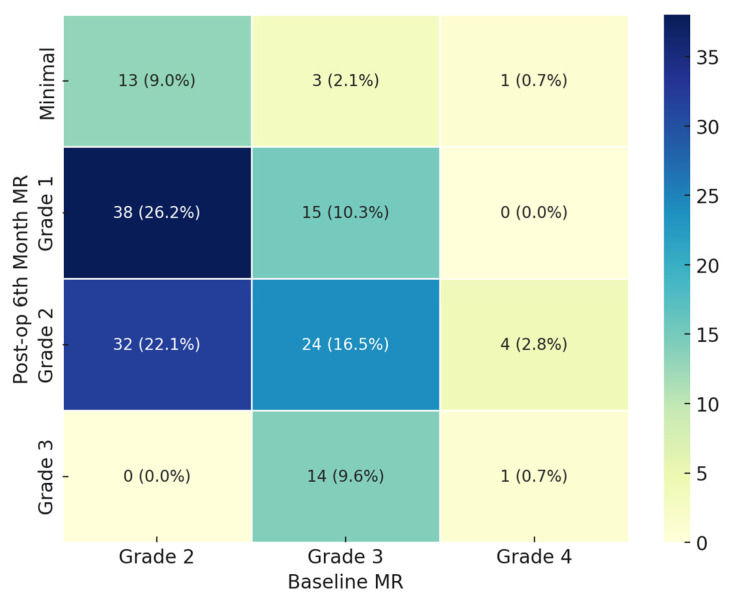
Heat map of the frequency distributions of mitral regurgitancy severity grades at baseline and at the 6-month follow-up.

**Table 1 medicina-60-02077-t001:** Demographic and clinical characteristics of the cases according to the grade of mitral regurgitation at baseline.

	MR 2 (*n* = 87)	MR 3–4 (*n* = 68)	*p*-Value	Overall (*n* = 155)
Age	78.0 (71.0–82.0)	81.0 (75.3–84.8)	**0.018** ^a^	79.0 (73.0–83.0)
Gender			0.230 ^b^	
Male	39 (44.8%)	24 (35.3%)		63 (40.6%)
Female	48 (55.2%)	44 (64.7%)		92 (59.4%)
BMI	26.5 ± 5.0	27.1 ± 5.0	0.488 ^c^	26.7 ± 5.0
NYHA			0.619 ^b^	
II	19 (21.8%)	19 (27.9%)		38 (24.5%)
III	55 (63.2%)	38 (55.9%)		93 (60.0%)
IV	13 (15.0%)	11 (16.2%)		24 (15.5%)
STS score	6.3 (4.1–9.4)	6.6 (4.4–10.3)	0.374 ^a^	6.5 (4.3–10.0)
CAD			0.238 ^b^	
Normal	32 (36.8%)	24 (35.3%)		56 (36.1%)
Non-obstructive	39 (44.8%)	24 (35.3%)		63 (40.7%)
Obstructive	16 (18.4%)	20 (20.4%)		36 (23.2%)
CABG	18 (20.7%)	12 (17.6%)	0.796 ^d^	30 (19.4%)
COPD	46 (52.9%)	31 (45.6%)	0.368 ^b^	77 (49.7%)
CVA	7 (8.0%)	6 (8.8%)	>0.999 ^e^	13 (8.4%)
PAD	6 (6.9%)	7 (10.3%)	0.642 ^d^	13 (8.4%)
DM	21 (24.1%)	18 (26.5%)	0.884 ^d^	39 (25.2%)
HT	70 (80.5%)	55 (80.9%)	>0.999 ^d^	125 (80.6%)
AF	24 (27.6%)	21 (30.9%)	0.787 ^d^	45 (29.0%)
Baseline GFR	65.2 ± 20.8	62.7 ± 18.8	0.450 ^c^	64.1 ± 20.0
Bicuspid Aorta	12 (13.8%)	8 (11.8%)	0.895 ^d^	20 (12.9%)
Baseline EF	55.0 (40.0–60.0)	47.5 (30.0–60.0)	**0.026** ^a^	50.0 (35.0–60.0)
Baseline LVEDD	4.8 (4.3–5.2)	4.9 (4.5–5.4)	**0.037** ^a^	4.8 (4.5–5.3)
Baseline LVESD	3.1 (2.6–3.8)	3.4 (2.8–4.2)	0.139 ^a^	3.2 (2.7–4.0)
Baseline septum thickness	1.3 (1.2–1.5)	1.3 (1.2–1.5)	0.871 ^a^	1.3 (1.2–1.5)
Baseline posterior wall thickness	1.3 (1.2–1.4)	1.3 (1.2–1.4)	0.883 ^a^	1.3 (1.2–1.4)
RWT	0.52 (0.46–0.62)	0.53 (0.46–0.59)	0.294 ^a^	0.52 (0.46–0.61)
LVMI	176.4 (141.8–201.2)	186.6 (163.9–218.7)	0.085 ^a^	179.8 (154.4–208.5)
Baseline LA diameter	4.6 (4.3–5.0)	4.8 (4.5–5.3)	**0.042** ^a^	4.6 (4.4–5.1)
Baseline AVMG	48.0 (42.0–63.0)	45.5 (41.0–56.5)	0.181 ^a^	47.5 (41.0–61.0)
Baseline sPAB	50.0 (35.0–61.0)	50.0 (42.0–65.0)	0.121 ^a^	50.0 (40.0–65.0)
MR Etiology				
Organic Functional	30 (34.4%) 57 (65.6%)	13 (19.1%) 55 (80.9%)	0.298	43 (27.7%) 112 (72.3%)

MR; mitral regurgitancy, BMI; body mass index, NYHA; New York Heart Association, STS; Society of Thoracic Surgeons, CAD; coronary artery disease, CABG; coronary artery bypass graft, COPD; chronic obstructive pulmonary disease, CVA; cerebro-vascular accident, PAD; peripheric artery disease, DM; diabetes mellitus, HT; hypertension, AF; atrial fibrillation, GFR; Glomerular filtration rate, EF; ejection fraction, LVEDD; left ventricular end-diastolic dimension, LVESD; left ventricular end-systolic dimension, RWT; relative wall thickness, LVMI; left ventricular mass index, LA; left atrial, AVMG; Aortic valve mean gradient, sPAB; systolic pulmonary artery pressure. Descriptive statistics for continuous variables were shown as mean ± SD or median (25th–75th) percentile, where appropriate. ^a^ Mann Whitney U test, ^b^ Pearson’s χ^2^ test, ^c^ Student’s *t*-test, ^d^ Continuity corrected χ^2^ test, ^e^ Fisher’s exact test.

**Table 2 medicina-60-02077-t002:** Other clinical characteristics of the cases according to the grade of mitral regurgitation at baseline.

	MRG 2 (*n* = 87)	MRG 3–4 (*n* = 68)	*p*-Value	Overall (*n* = 155)
Access site			0.700 ^a^	
Trans-axillary	3 (3.4%)	4 (5.9%)		7 (4.5%)
Trans-femoral	84 (96.6%)	64 (94.1%)		148 (95.5%)
Pre-dilation	68 (78.2%)	44 (64.7%)	0.094 ^b^	112 (72.3%)
Post-dilation	9 (10.3%)	3 (4.4%)	0.285 ^b^	12 (7.7%)
Valve type			0.149 ^c^	
SAPIEN XT	80 (92.0%)	58 (85.2%)		138 (89.0%)
Edwards SAPIEN 3	5 (5.8%)	5 (7.4%)		10 (6.5%)
LOTUS	1 (1.1%)	5 (7.4%)		6 (3.9%)
ACURATE neo	1 (1.1%)	0 (0.0%)		1 (0.6%)
AS Group			0.261 ^c^	
HG	50 (58.1%)	41 (60.3%)		91 (59.1%)
LFLG	6 (7.0%)	10 (14.7%)		16 (10.4%)
Paradoxical LFLG	1 (1.2%)	1 (1.5%)		2 (1.3%)
VSAS	29 (33.7%)	16 (23.5%)		45 (28.2%)

AS; aortic stenosis, HG; high gradient, LFLG; low flow low gradient, VSAS; very severe aortic stenosis. ^a^ Fisher’s exact test, ^b^ Continuity corrected χ^2^ test, ^c^ Fisher Freeman Halton test.

**Table 3 medicina-60-02077-t003:** Demographic and clinical characteristics of the patients according to the improvement in mitral regurgitation grade at 6-month follow-up.

	No Change (*n* = 46)	Improvement = 1 Grade (*n* = 63)	Improvement ≥ 2 Grade (*n* = 36)	*p*-Value
Age	77.5 (72.8–82.3)	79.0 (74.0–84.0)	79.0 (70.3–82.8)	0.360 ^a^
Gender				0.342 ^b^
Male	22 (47.8%)	23 (36.5%)	12 (33.3%)	
Female	24 (52.2%)	40 (63.5%)	24 (66.7%)	
BMI	26.4 ± 5.4	26.5 ± 4.5	27.3 ± 5.7	0.711 ^c^
NYHA				0.500 ^b^
II	15 (32.6%)	13 (20.6%)	8 (22.2%)	
III	26 (56.5%)	42 (66.7%)	21 (58.3%)	
IV	5 (10.9%)	8 (12.7%)	7 (19.5%)	
STS score	6.5 (4.4–10.2)	7.3 (4.4–10.3)	6.4 (4.0–9.5)	0.559 ^a^
CAD				0.602 ^b^
Normal	16 (34.8%)	24 (38.1%)	13 (36.1%)	
Non-obstructive	22 (47.8%)	24 (38.1%)	12 (33.3%)	
Obstructive	8 (17.4%)	15 (23.8%)	11 (30.6%)	
CABG	10 (21.7%)	13 (20.6%)	5 (13.9%)	0.630 ^b^
COPD	28 (60.9%)	27 (42.9%)	16 (44.4%)	0.146 ^b^
CVA	4 (8.7%)	4 (6.3%)	4 (11.1%)	0.640 ^d^
PAD	1 (2.2%)	7 (11.1%)	5 (13.9%)	0.108 ^d^
DM	9 (19.6%)	13 (20.6%)	14 (38.9%)	0.078 ^b^
HT	39 (84.8%)	48 (76.2%)	30 (83.3%)	0.478 ^b^
AF	14 (30.4%)	20 (31.7%)	6 (16.7%)	0.237 ^b^
Baseline GFR	67.4 ± 20.3	63.0 ± 19.7	61.6 ± 20.4	0.382 ^c^
Bicuspid Aorta	4 (8.7%)	9 (14.3%)	7 (19.4%)	0.371 ^b^
Baseline EF	44.5 (35.0–55.0)	55.0 (35.0–60.0)	60.0 (36.3–65.0)	**0.012** ^a^
Baseline LVEDD	5.3 (4.6–5.8)	4.6 (4.3–5.0)	4.9 (4.4–5.2)	**<0.001** ^a^
Baseline LVESD	3.7 (3.1–4.3)	3.0 (2.5–3.5)	3.1 (2.6–3.9)	**<0.001** ^a^
Baseline septum thickness	1.3 (1.2–1.5)	1.3 (1.2–1.5)	1.3 (1.2–1.5)	0.710 ^a^
Baseline posterior wall thickness	1.3 (1.2–1.4)	1.3 (1.2–1.4)	1.3 (1.2–1.4)	0.949 ^a^
RWT	0.49 (0.42–0.59)	0.54 (0.49–0.60)	0.52 (0.48–0.59)	**0.026** ^a^
LVMI	206.4 (168.0–247.4)	177.6 (145.4–189.3)	171.5 (154.9–192.1)	**<0.001** ^a^
Baseline LA diameter	4.8 (4.6–5.6)	4.6 (4.3–4.9)	4.6 (4.3–5.1)	**0.018** ^a^
Baseline AVMG	45.0 (41.0–62.0)	45.0 (41.0–61.0)	49.0 (42.0–61.8)	0.662 ^a^
Baseline sPAB	52.5 (35.0–66.3)	50.0 (35.0–60.0)	45.0 (40.0–60.0)	0.631 ^a^
MR Aetiology				
Organic Functional	11 (23.9%) 35 (76.1%)	15 (23.8%) 48 (76.2%)	13 (36.1%) 23 (63.9%)	0.283

MR; mitral regurgitancy, BMI; body mass index, NYHA; New York Heart Association, STS; Society of Thoracic Surgeons, CAD; coronary artery disease, CABG; coronary artery bypass graft, COPD; chronic obstructive pulmonary disease, CVA; cerebro-vascular accident, PAD; peripheric artery disease, DM; diabetes mellitus, HT; hypertension, AF; atrial fibrillation, GFR; Glomerular filtration rate, EF; ejection fraction, LVEDD; left ventricular end-diastolic dimension, LVESD; left ventricular end-systolic dimension, RWT; relative wall thickness, LVMI; left ventricular mass index, LA; left atrial, AVMG; Aortic valve mean gradient, sPAB; systolic pulmonary artery pressure. Descriptive statistics for continuous variables were shown as mean ± SD or median (25th–75th) percentile, where appropriate. ^a^ Kruskal–Wallis test, ^b^ Pearson’s χ^2^ test, ^c^ One-Way ANOVA, ^d^ Fisher–Freeman Halton test. There was no statistically significant difference between the groups indicated by the same uppercase letters (*p* > 0.05).

**Table 4 medicina-60-02077-t004:** Other clinical characteristics of the patients according to the improvement in mitral regurgitation grade at 6-month follow-up.

	No Change (*n* = 46)	Improvement = 1 Grade (*n* = 63)	Improvement ≥ 2 Grade (*n* = 36)	*p*-Value
Access site				0.194 ^a^
Trans-axillary	4 (8.7%)	1 (1.6%)	2 (5.6%)	
Trans-femoral	42 (91.3%)	62 (98.4%)	34 (94.4%)	
Pre-dilation	28 (60.9%)	50 (79.4%)	27 (75.0%)	0.095 ^b^
Post-dilation	3 (6.5%)	5 (7.9%)	2 (5.6%)	>0.999 ^a^
Valve type				0.915 ^a^
SAPIEN XT	41 (89.1%)	55 (87.3%)	32 (88.9%)	
Edwards SAPIEN 3	3 (6.5%)	5 (7.9%)	2 (5.6%)	
LOTUS	1 (2.2%)	3 (4.8%)	2 (5.6%)	
ACURATE neo	1 (2.2%)	0 (0.0%)	0 (0.0%)	
AS Group				0.917
HG	25 (55.6%)	39 (61.9%)	21 (58.3%)	
LFLG	5 (11.1%)	6 (9.5%)	3 (8.3%)	
Paradoxical LFLG	1 (2.2%)	0 (0.0%)	1 (2.8%)	
VSAS	14 (31.1%)	18 (28.6%)	11 (30.6%)	

AS; aortic stenosis, HG; high gradient, LFLG; low flow low gradient, VSAS; very severe aortic stenosis. ^a^ Fisher–Freeman Halton test, ^b^ Pearson’s χ^2^ test.

**Table 5 medicina-60-02077-t005:** Determining the best predictors that affect the improvement in mitral regurgitation grade at 6-month follow-up—the results of multinominal logistic regression analysis via Backward elimination procedure.

	OR (95% CI)	Wald	*p*-Value
Improvement = 1 grade			
Baseline MR 3–4	4.098 (1.409–11.919)	6.702	**0.010**
Baseline EF *	0.981 (0.634–1.519)	0.007	0.933
Baseline LVEDD **	1.291 (1.097–1.520)	9.408	**0.002**
Baseline LVMI **	0.945 (0.825–1.083)	0.660	0.417
Baseline LA **	1.124 (1.030–1.227)	6.816	**0.009**
Improvement ≥2 grade			
Baseline MR 3–4	13.514 (4.013–45.508)	17.665	**<0.001**
Baseline EF *	1.701 (1.007–2.871)	3.950	**0.047**
Baseline LVEDD **	1.345 (1.112–1.628)	9.296	**0.002**
Baseline LVMI **	1.109 (1.020–1.207)	5.835	**0.024**
Baseline LA **	1.121 (1.015–1.237)	5.069	**0.024**

MR: mitral regurgitation; EF: ejection fraction; LVEDD: left ventricular end-diastolic gradient; LVMI: left ventricular mass index; LA: left atrium. No change in mitral regurgitation grade was considered as the reference group. OR: Odds ratio, CI: Confidence interval, * Impact of each 10-unit increase on improvement, ** Impact of each 0.1-unit decrease on improvement.

**Table 6 medicina-60-02077-t006:** The association between baseline mitral regurgitation grade and the improvement in mitral regurgitation grade at 6-month follow-up with one-year mortality—the results of univariate logistic regression analysis.

	Alive	Exitus	*p*-Value	OR (95% CI)
Baseline MR				
2	80 (60.2%)	7 (31.8%)	-	1.000
3–4	53 (39.8%)	15 (68.2%)	**0.017**	3.235 (1.236–8.464)
Improvement in MR (6-month follow-up)				
≥2 grade	32 (24.1%)	4 (33.3%)	-	1.000
1 grade	58 (43.6%)	5 (41.7%)	0.599	0.690 (0.173–2.752)
No change	43 (32.3%)	3 (25.0%)	0.465	0.558 (0.117–2.670)

MR: Mitral regurgitation. OR: Odds ratio, CI: Confidence interval.

## Data Availability

The data presented in this study are available on request from the corresponding author as most of the data is derived from government social security system.

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
