# Peer review of "Prognostic Implications and Predictors of Mitral Regurgitancy Reduction After Transcatheter Aortic Valve Implantation"

_medicina, 2024, doi:10.3390/medicina60122077_

Round 1

Reviewer 1 Report

Comments and Suggestions for Authors

The study investigates the predictors and prognostic implications of mitral regurgitation reduction following transcatheter aortic valve implantation in 156 patients with moderate to severe MR. It identifies baseline echocardiographic parameters, including lower left ventricular end-diastolic dimension, left atrial size, left ventricular mass index, and higher ejection fraction, as significant predictors of MR improvement, while MR reduction did not independently predict one-year mortality. The findings highlight the need for further research to optimize patient selection and management strategies in TAVI candidates with concomitant MR.

To rule out selection bias, it would be valuable to compare the baseline characteristics of excluded patients (if data are available) with those of the included cohort.

The manuscript mentions that MR was evaluated using multiple methods, but a more detailed explanation of how these methods were prioritized or weighted would provide better insight into how MR grades were assigned.

The authors should include sensitivity analyses to evaluate the robustness of their findings.

The results section does not clearly differentiate between significant and non-significant outcomes. Consistent reporting of exact p-values would enhance the transparency of the findings.

Effect sizes for continuous variables, such as odds ratios for baseline LVEDD, should be accompanied by confidence intervals to better represent the precision of these estimates.

Although the manuscript discusses functional versus organic MR, it lacks a subgroup analysis to compare these two etiologies, which could reveal clinically meaningful differences.

While the discussion integrates the findings with prior studies, it overlooks the opportunity to elaborate on the innovative role of LVMI as a predictor of MR reduction. The authors should emphasize why this finding is noteworthy compared to established predictors like EF and LVEDD. Moreover authors should expand the background section of their manuscript to include a discussion of the potential risks associated with percutaneous procedures, despite their numerous advantages over traditional cardiac surgery ( doi: 10.1016/j.hrthm.2024.06.030. ,  doi: 10.1055/s-0042-1758073. )

Comments on the Quality of English Language

authors should reduce the overall complexity of their sentences to improve the readibility of their manuscript.

Author Response

Comment 1: To rule out selection bias, it would be valuable to compare the baseline characteristics of excluded patients (if data are available) with those of the included cohort.

Response 1: 

We appreciate the comment regarding the potential for selection bias and the suggestion to compare the baseline characteristics of excluded patients. Both participating centers in this study are tertiary reference centers for valvular heart diseases, and a significant proportion of our patients are referred from other cities. As a result, many of these patients continue their follow-up care at their local hospitals, limiting our access to comprehensive follow-up data.

To ensure the reliability of our findings and reduce the impact of incomplete data, we performed our analysis exclusively on patients with complete follow-up records. While we acknowledge the inherent limitation of potentially introducing selection bias, this approach ensures that our analysis is based on high-quality, consistent data. We believe this strengthens the validity of our conclusions.  As excluding this group of patients is primarily due to a lack of follow-up data, we believe this approach minimizes any significant risk of bias. However, if the reviewer deems it essential, we are willing to attempt a comparison of available baseline characteristics between included and excluded patients as part of  major revision. We added the sentence regarding the high amount of excluded patient in manuscripts methods section as highlighted. To clarify, we have added information specifying that out of 615 patients who underwent the TAVI procedure, 206 were identified as having moderate or severe MR. Of these, 156 patients had complete follow-up data and were included in the analysis. We apologize for any confusion caused by the original manuscript.

Comment 2: The manuscript mentions that MR was evaluated using multiple methods, but a more detailed explanation of how these methods were prioritized or weighted would provide better insight into how MR grades were assigned

Response 2: 

In our study, MR severity was determined using  semiquantitative measures, such as the width of the vena contracta, and quantitative methods, including the calculation of the effective regurgitant orifice area. These methods were applied in accordance with current echocardiographic guidelines, ensuring robust and reliable assessment.

Echocardiographic examinations were performed by two highly experienced echocardiographers, both certified by national boards and the European Association of Cardiovascular Imaging (EACVI). The choice between semiquantitative and quantitative methods was guided by image quality and the feasibility of obtaining precise measurements. 

The related sentence is added to methods section as highlighted.

Comment 3: The authors should include sensitivity analyses to evaluate the robustness of their findings

Response 3: We performed detailed sensitivity analyses based on your comment and presented them as a supplementary file attached  on notes section. As there is substantial information, we have not included these details in the revised manuscript but are prepared to make additions based on your further recommendations.

Comment 4:The results section does not clearly differentiate between significant and non-significant outcomes. Consistent reporting of exact p-values would enhance the transparency of the findings

Response 4: We revised the results section to ensure a clear distinction between significant and non-significant outcomes.  The corrections have been highlighted in the revised manuscript for your review.

Comment 5: Effect sizes for continuous variables, such as odds ratios for baseline LVEDD, should be accompanied by confidence intervals to better represent the precision of these estimates

Response 5: Confidence intervals for variables evaluated  in multinominal logistic regression analysis were included in table 5.

Comment 6: Although the manuscript discusses functional versus organic MR, it lacks a subgroup analysis to compare these two etiologies, which could reveal clinically meaningful differences

 Response 6: 

Thank you for highlighting the importance of a subgroup analysis comparing organic and functional MR etiologies. While we agree that such an analysis could reveal clinically meaningful differences, the primary objective of this study was not to investigate these differences. Additionally, there is a significant imbalance in patient numbers between the two groups, as patients with organic MR were more frequently referred for surgery. This imbalance introduces potential bias, limiting the reliability of a subgroup analysis within the current dataset.

Your comments have encouraged us to consider a more focused investigation of organic versus functional MR in a carefully selected patient cohort. We plan to conduct a dedicated analysis in the future to explore this topic more comprehensively.

We added a highlighted sentece regarding this topic to limitations section.

Comment 7: While the discussion integrates the findings with prior studies, it overlooks the opportunity to elaborate on the innovative role of LVMI as a predictor of MR reduction. The authors should emphasize why this finding is noteworthy compared to established predictors like EF and LVEDD.

Response 7: We edited the regarding paragraph in discussion section with additional references as you suggested and highlighted it.

Comment 8: Moreover authors should expand the background section of their manuscript to include a discussion of the potential risks associated with percutaneous procedures, despite their numerous advantages over traditional cardiac surgery

Response 8: We have addressed the comment by incorporating the requested changes into the manuscript, which are highlighted accordingly, and have included the suggested reference.

Comment 9: Authors should reduce the overall complexity of their sentences to improve the readibility of their manuscript.

Response 9: We have carefully reviewed the manuscript and simplified complex sentences to enhance readability and clarity. We hope this improves the accessibility and overall quality of the paper.

Reviewer 2 Report

Comments and Suggestions for Authors

This study aimed to assess the prognostic significance of baseline mitral MR and its improvement following TAVI, as well as to identify factors related to MR reduction post-TAVI. Key findings showed that 68% of patients experienced a reduction in MR severity at 6 months, increasing to 81% at 1 year. Factors associated with greater MR reduction included baseline LVEF, LVEDD, LVMI, and LA size. However, the reduction in MR severity did not independently predict 1-year survival outcome.

Major concern

Uncertainty in Prognostic Prediction
The study did not show a significant impact of MR improvement following TAVI on 1-year survival. As a result, factors associated with MR improvement (such as LVEF) lost their predictive value. Further investigation is needed to understand why the results differ from those of other studies and to address this gap in the literature regarding the effect of MR improvement on survival.

Minor concern

1.      As a retrospective cohort study, there is a potential for selection bias that may have influenced the results. This bias could arise from the way patients were selected for inclusion, potentially affecting the reliability of the findings and limiting the generalizability of the conclusions.

2.      The study found no significant difference in MR outcomes between patients with primary MR and secondary MR. This lack of distinction may be due to the relatively small number of patients with primary MR in this cohort, possibly leading to a selection bias, as patients with primary MR and AS may have been more likely to undergo combined valve surgery rather than TAVI.

3.      The study excluded patients with missing follow-up data, and the influence of this missing data on the results is unclear. The absence of detailed analyses on how missing data may have affected the findings limits the interpretability and generalizability of the results.

4.      I would like the prognostic data in Table 6 to be clearly shown as Figure with Kaplan-Meier curves.

5.      There are a few spelling errors that need to be corrected.

Author Response

Comment 1 : The study did not show a significant impact of MR improvement following TAVI on 1-year survival. As a result, factors associated with MR improvement (such as LVEF) lost their predictive value. Further investigation is needed to understand why the results differ from those of other studies and to address this gap in the literature regarding the effect of MR improvement on survival.

Response 1:We appreciate the reviewer’s insightful comment regarding the relationship between MR reduction, mortality, and the importance of MR reduction predictors. In response, we have added an explanation to the discussion section to address this concern, emphasizing the observed non-significant trend toward reduced mortality following MR reduction and referencing literature findings that report a significant relationship between MR reduction and mortality. This provides a reasonable explanation for the clinical relevance of identifying predictors of MR reduction, despite the absence of a significant association with mortality in our study. These changes have been highlighted in the revised manuscript for clarity. 

Comment 2:   As a retrospective cohort study, there is a potential for selection bias that may have influenced the results. This bias could arise from the way patients were selected for inclusion, potentially affecting the reliability of the findings and limiting the generalizability of the conclusions

Response 2: 

We acknowledge that selection bias is an inherent limitation of retrospective analyses. To mitigate this, we included only patients with complete follow-up data from two high-volume tertiary centers specializing in valvular heart disease, ensuring a well-defined and representative cohort. However, we recognize that this approach may limit generalizability, as patients with incomplete follow-up data were excluded. We have expanded the limitations section of the manuscript to address this concern and emphasize the need for future prospective studies to validate our findings. The revisions have been highlighted in the updated manuscript.

Comment 3: The study found no significant difference in MR outcomes between patients with primary MR and secondary MR. This lack of distinction may be due to the relatively small number of patients with primary MR in this cohort, possibly leading to a selection bias, as patients with primary MR and AS may have been more likely to undergo combined valve surgery rather than TAVI.

Response 3: 

Thank you for highlighting the importance of a subgroup analysis comparing organic and functional MR etiologies. While we agree that such an analysis could reveal clinically meaningful differences, the primary objective of this study was not to investigate these differences. Additionally, there is a significant imbalance in patient numbers between the two groups, as patients with organic MR were more frequently referred for surgery. This imbalance introduces potential bias, limiting the reliability of a subgroup analysis within the current dataset.

Your comments have encouraged us to consider a more focused investigation of organic versus functional MR in a carefully selected patient cohort. We plan to conduct a dedicated analysis in the future to explore this topic more comprehensively.

We added a highlighted sentence regarding this topic to limitations section.

Comment 4: The study excluded patients with missing follow-up data, and the influence of this missing data on the results is unclear. The absence of detailed analyses on how missing data may have affected the findings limits the interpretability and generalizability of the results.

Response 4: To ensure the reliability of our findings and reduce the impact of incomplete data, we performed our analysis exclusively on patients with complete follow-up records. While we acknowledge the inherent limitation of potentially introducing selection bias, this approach ensures that our analysis is based on high-quality, consistent data. We believe this strengthens the validity of our conclusions.  As excluding this group of patients is primarily due to a lack of follow-up data, we believe this approach minimizes any significant risk of bias. However, if the reviewer deems it essential, we are willing to attempt a comparison of available baseline characteristics between included and excluded patients as part of  major revision. We added the sentence regarding the high amount of excluded patient in manuscripts methods section as highlighted. 

Comment 5:    I would like the prognostic data in Table 6 to be clearly shown as Figure with Kaplan-Meier curves.

Response 5: While we recognize the value of Kaplan-Meier curves in illustrating survival data, the mortality data in our study were derived from the national health services database. Due to access limitations, we could only obtain mortality data in 6-month intervals, without exact dates of death. This constraint makes Kaplan-Meier analysis inaccurate, as the precise timing of events is necessary for reliable curve generation. We have  retained the data presentation in tabular form to ensure accuracy and transparency

Comment 6: There are a few spelling errors that need to be corrected

Response 6: Thank you for pointing this out. We have thoroughly reviewed the manuscript for spelling errors and corrected any identified mistakes

Round 2

Reviewer 1 Report

Comments and Suggestions for Authors

Congratulations to the authors for their effort in improving their manuscript.

Reviewer 2 Report

Comments and Suggestions for Authors

The manuscript has been sufficiently corrected and acceptable to publish.

I would have liked to see a Kaplan-Meier analysis of the event rate, but I understand that this cannot be made public.